# META-REFERENTIAL GAMES TO LEARN COMPOSITIONAL LEARNING BEHAVIOURS

## ABSTRACT

Referring to *compositional learning behaviours* as the ability to learn to generalise compositionally from a limited set of stimuli, that are combinations of supportive stimulus components, to a larger set of novel stimuli, i.e. novel combinations of those same stimulus components, we acknowledge compositional learning behaviours as a valuable feat of intelligence that human beings often rely on, and assume their collaborative partners to use similarly. In order to build artificial agents able to collaborate with human beings, we propose a novel benchmark to investigate state-of-the-art artificial agents abilities to exhibit compositional learning behaviours. We provide baseline results on the single-agent tasks of learning compositional learning behaviours, using state-of-the-art RL agents, and show that our proposed benchmark is a compelling challenge that we hope will spur the research community towards developing more capable artificial agents.

## 1 INTRODUCTION

Framing the ability to learn to generalise compositionally from a limited set of combinations of supportive stimulus components to a larger set of novel combinations of those same supportive stimulus components as *compositional learning behaviours*, we acknowledge compositional learning behaviours as a valuable feat of intelligence that human beings often rely on, and assume their collaborative partners to use similarly.

In order to build artificial agents able to collaborate with human beings, it is important to endow the former with similar abilities. Thus, we propose to investigate state-of-the-art artificial agents abilities to exhibit compositional learning behaviours.

**Compositional Language Emergence as a Proxy for *Compositional Behaviours*.** The field of language emergence raises the question of how to make artificial languages emerge with similar properties to natural languages, with compositionality at the forefront of those properties(Lazaridou et al., 2018; Baroni, 2019; Guo et al., 2019; Li & Bowling, 2019; Ren et al., 2020). Referential game (Lewis, 1969) variants are the primary tools used to make language emerge between a pair or population of artificial agents (Denamganaï & Walker, 2020a). In this context, it has been shown that emerging languages are far from being 'natural-like' protolanguages (Kottur et al., 2017; Chaabouni et al., 2019a;b), but sufficient conditions can be found to further the emergence of compositional languages and generalising learned representations (e.g. (Kottur et al., 2017; Lazaridou et al., 2018; Choi et al., 2018; Bogin et al., 2018; Guo et al., 2019; Korbak et al., 2019; Chaabouni et al., 2020; Denamganaï & Walker, 2020b)). Nevertheless, the ability of deep-learning agents to generalise compositionally in a systematic fashion has been called into question, especially when it comes to language grounding in general (Hill et al., 2019b), on relational reasoning tasks (Bahdanau et al., 2019), or on the SCAN benchmark (Lake & Baroni, 2018; Loula et al., 2018; Liška et al., 2018), and more recently the gSCAN benchmark (Ruis et al., 2020). Neural networks induction biases have been investigated towards finding necessary conditions that favour the emergence of compositional generalisation/systematicity (Hill et al., 2019b; Słowik et al., 2020; Korrel et al., 2019; Lake, 2019; Russin et al., 2019).

Chaabouni et al. (2020) showed that, when a specific kind of compositionality is found in the emerging languages (the kind that scores high on the positional disentanglement (posdis) metric for compositionality that they proposed), then it is a sufficient condition for systematicity to emerge. And, more importantly, they showed that emergence of a compositional language (in the sense

of any of the existing compositionality-testing metric, e.g. topographic similarity (Brighton & Kirby, 2006)) is not a necessary condition for systematic generalisation (as evaluated by a zero-shot compositional learning test) as non-compositional, emerging languages have been shown to support systematicity of the pair of agents wielding them in a (generative) referential game.

In this work, we focus directly on systematic generalisation/systematicity as a behaviour, as success in our proposed task is directly related to the agents ability to exhibit compositional learning behaviours, thus investigating whether they can generalise compositionally in a systematic and online fashion (see Section 4).

Moreover, contrary to our framework, when studying compositionality in the context of any referential game variant instantiated with a given dataset of stimuli (of any nature, e.g. one-hot-encoded (Kottur et al., 2017; Chaabouni et al., 2020; Ren et al., 2020; Resnick et al., 2019), visual (Lazaridou et al., 2018; Choi et al., 2018), or multi-modal (Evtimova et al., 2017)), then the trained pair of agents only exhibits *compositional behaviours* for said dataset within the constraint of the training, thus failing short of learning compositional behaviours that generalises to novel situations/stimuli/datasets. In other words, agents are failing to learn *compositional learning behaviours*. Our framework aims to remedy this gap via, firstly, a meta-learning framing of the task and, secondly, via the introduction of a novel stimulus representation that compromises between the one-hot-encoded representation of most symbolic stimuli benchmarks and the continuous representations akin to latent embeddings as found in the unsupervised learning approaches, without sacrificing the symbolic aspects (see Section 3).

**Meta-Referential Games for *Compositional Learning Behaviours*.** Learning compositional learning behaviours pertains to meta-learning. In order to frame referential games in a meta-learning context where successful agents are able to generalise their abilities to exhibit compositional behaviours onto novel situations/datasets, it is necessary to provide the agent at training-time with a distribution of tasks/stimuli/datasets that encompasses the kind of tasks/stimuli/datasets that the agent will experience after being deployed.

For instance, using a set of datasets with different generative factors/attributes as our distribution of tasks would force us to focus on only one modality shared among all datasets, for instance the visual modality. Doing so would add to our work the assumption of working with visual stimuli only, which is quite constraining. Instead, we choose to assume that stimuli are disentangled, independently of their nature. Thus, using one-hot-encoded (OHE) vectors, that are disentangled abstract stimuli, would fit the bill with regards to our assumption and our work would retain some external validity. It would be applicable whenever stimuli can be discretely categorised over a set of generative factors/attributes **with discrete values**.

Further more, once a semantic structure is chosen, i.e. a number of generative factors/attributes, $N_{dim}$, and a number of possible values for each $(d(i))_{i \in [1;N_{dim}]}$, **it is impossible to extend our meta-distribution to differently semantically structured stimuli without changing the shape of our OHE vectors,** $d_{stim} = \sum_{i \in [1;N_{dim}]} d(i)$. Most state-of-the-art deep learning architectures rely on the assumption that the dimensions of their input spaces are constant. Thus, in order to provide a meta-distribution over differently semantically structured stimuli spaces, we propose a different representation, entitled **symbolic continuous stimulus representation** (SCS), it is detailed in Section 3. Using this SCS representation, we can meta-train agents in a meta-referential game settings where the semantic structure observed can be randomised over without changing the shape of the stimulus space and without the overly constraining assumption that the representation must be discrete.

Our contributions are threefold:

- We propose the Symbolic Continuous Stimulus (SCS) representation as a versatile encoding of symbolic spaces, which, on the contrary to one-hot-encoded representation, allows for infinitely many semantic structure to be instantiated without changing the shape of the representation, and rely on continuous values that are more akin to the kind of stimuli found in the real world, as opposed to discrete valued one-hot-encodings.

- We cast the problem of learning compositional behaviours as a meta-reinforcement learning problem, using (discriminative) referential games, containing a meta-reinforcement learn-

ing formulation of the binding problem, thus allowing it to be studied under the lens of state-of-the-art RL algorithms.

- We provide baseline results on the single-agent task of learning compositional learning behaviours, using state-of-the-art RL agents, and show that our proposed benchmark is a compelling challenge that we hope will spur the research community towards developing more capable artificial agents.

## 2 BACKGROUND & RELATED WORKS

Referential games are at an interface between the language processing subfields of language emergence, language grounding, and, when not using on-hot-encoded stimuli, the computer vision subfield of unsupervised representation learning. While language emergence raises the question of how to make artificial languages emerge with similar properties to natural languages, or at least 'natural-like' protolanguages, with compositionality at the forefront of those properties(Baroni, 2019; Guo et al., 2019; Li & Bowling, 2019; Ren et al., 2020), language grounding is concerned with the ability to ground the meaning of (natural) language utterances into some sensory processes, with the visual modality being the main focus of research. On one hand, emerging artificial languages' compositionality has been shown to further the learnability of said languages (Kirby, 2002; Smith et al., 2003; Brighton, 2002; Li & Bowling, 2019) and, on the other hand, natural languages' compositionality promises to increase the generalisation ability of the artificial agent that would be able to rely on them as a grounding signal, as it has been found to produce learned representations that generalise, when measured in terms of the data-efficiency of subsequent transfer and/or curriculum learning (Higgins et al., 2017; Mordatch & Abbeel; Moritz Hermann et al.; Jiang et al., 2019). More in touch with the current context of this study, Chaabouni et al. (2020) showed that, when a specific kind of compositionality is found in the emerging languages (the kind that scores high on the positional disentanglement (posdis) metric for compositionality that they proposed), then it is a sufficient condition for systematicity to emerge.

**Language Compositionality & Compositional Systematic Generalisation/Systematicity.** As a concept, compositionality has been the focus of many definition attempts. For instance, it can be defined as "the algebraic capacity to understand and produce novel combinations from known components"(Loula et al. (2018) referring to Montague (1970)) or as the property according to which "the meaning of a complex expression is a function of the meaning of its immediate syntactic parts and the way in which they are combined" (Krifka, 2001). Although difficult to define, the commmunity seem to agree on the fact that it would enable learning agents to exhibit systematic generalisation abilities (also referred to as combinatorial generalisation (Battaglia et al.)). Some of the ambiguities that come with those loose definitions start to be better understood and explained, as in the work of Hupkes et al. (2019). While often studied in relation to languages, it is usually defined with a focus on behaviours. In this paper, we refer to compositional behaviours as "the ability to entertain a given thought implies the ability to entertain thoughts with semantically related contents"(Fodor et al., 1988), and thus use it interchangeably with systematicity, following the classification made by Hupkes et al. (2019).

Compositionality, as a property of languages, will be referred to as linguistic compositionality, in this paper. It can be difficult to measure. Brighton & Kirby (2006)'s *topographic similarity* (**topsim**) which is acknowledged by the research community as the main quantitative metric for compositionality (Lazaridou et al., 2018; Guo et al., 2019; Słowik et al., 2020; Chaabouni et al., 2020; Ren et al., 2020). Recently, taking inspiration from disentanglement metrics, Chaabouni et al. (2020) proposed two new metrics entitled **posdis** (positional disentanglement metric) and **bosdis** (bag-of-symbols disentanglement metric), that have been shown to be differently 'opinionated' in the sense that they each seem to capture different ways in which a language can be shown to be (linguistically) compositional.

**Binding Problem & Meta-Learning.** Following Greff et al. (2020), we refer to the binding problem as the "inability to dynamically and flexibly bind information that is distributed throughout the network" of deep learning architectures. We note that relational responding ("adjusting the (task-specific) response to an object based on its relation to other objects" ; for instance, one way to facilitate it in a neural network "is to organise its internal information flow (i.e. computations) in

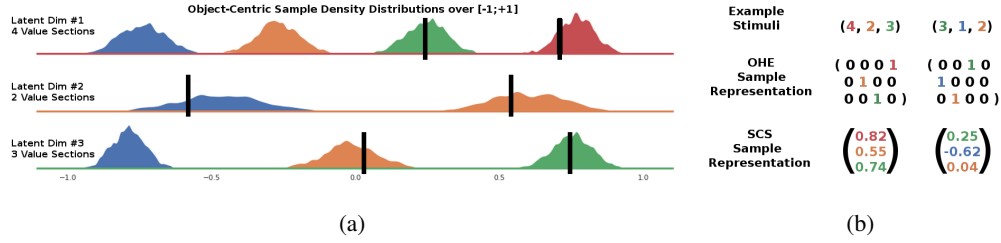

(a)                                                              (b)

Figure 1: **(a):** representation of the gaussian kernels corresponding to each value sections on each latent factor/attribute dimension, in the case of $N_{dim} = 3$. Black vertical bars indicate the $g_{l(i)}$ value samples from each gaussian kernels corresponding to the $l(i)$ values instantiated on each latent dimension $i$, to construct the SCS representations of the example stimuli in (b). **(b):** OHE and SCS representations of example stimuli for a semantic structure/symbolic space with $N_{dim} = 3$, $d(0) = 4$, $d(1) = 2$, $d(2) = 3$.

a way that reflects the graph structure of relations and objects") is central to solving the binding problem instantiated in our proposed benchmark.

Relational Frame Theory (RTF) distinguishes "two types of entailment that humans primarily use to derive (unobserved) relations: mutual entailment [(derive additional relations between two objects based on a given relation between them)] and combinatorial entailment [(derive new relations between two objects, based on their relations with a shared third object)]". Within those terms, the ability to perform (sequential) combinatorial entailment is at the center of the benchmark challenges.

## 3    SYMBOLIC CONTINUOUS STIMULUS REPRESENTATION

We introduce a symbolic continuous (as opposed to discrete) stimulus (SCS) representation which has the particularity of enabling the representation of stimuli sampled from differently semantically structured symbolic spaces while maintaining the same representation shape, as opposed to the one-hot encoded (OHE) representation. We will refer to this as the *representation shape invariance property* of the SCS representation.

Namely, defining the semantic structure of an $N_{dim}$-dimensioned symbolic space by the tuple $(d(i))_{i \in [1;N_{dim}]}$ where $N_{dim}$ is the number of factor dimensions, $d(i)$ is the number of values for each factor dimension $i$, then the representation shape of any stimulus sampled from any such $N_{dim}$-dimensioned symbolic space is a vector over $[-1, +1]^{N_{dim}}$. Note that this shape does not depend on the $d(i)$'s values, as opposed to the OHE representation which samples vectors from the discrete space $\{0, 1\}^{d_{OHE}}$ where $d_{OHE} = \Sigma_{i=1}^{N_{dim}} d(i)$.

From a given semantic structure, $(d(i))_{i \in [1;N_{dim}]}$, the representation sampling space is built as follows: for each factor dimension $i$, the $[-1, +1]$ range is partitioned in $d(i)$ value sections, each corresponding to one of the $d(i)$ symbolic values available on the $i$-th factor dimension. Sampling a stimulus from this symbolic space boils down to instantiating latent values $l(i)$ on each factor dimension $i$, such that $l(i) \in [1; d(i)]$. Differently from the OHE representation, the SCS representation of this stimulus is a continuous vector whose $i$-th dimension is populated with a sample from a corresponding gaussian distribution over the $l(i)$-th partition, $g_{l(i)} \sim \mathcal{N}(\mu_{l(i)}, \sigma_{l(i)})$, where $\mu_{l(i)}$ is the mean of the gaussian distribution, uniformly sampled to fall within the range of the $l(i)$-th partition, and $\sigma_{l(i)}$ is the standard deviation of the gaussian distribution, uniformly sampled over the range $[\frac{2}{12d(i)}, \frac{2}{6d(i)}]$. $\mu_{l(i)}$ and $\sigma_{l(i)}$ are sampled in order to guarantee (i) that the scale of the gaussian distribution is large enough, but (ii) not larger than the size of the partition section it should fit in. Figure 1a shows an example of such instantiation of the different gaussian distributions over each factor dimensions' $[-1, +1]$ range, and Figure 1b highlights how the two representations differ when representating the same example stimuli.

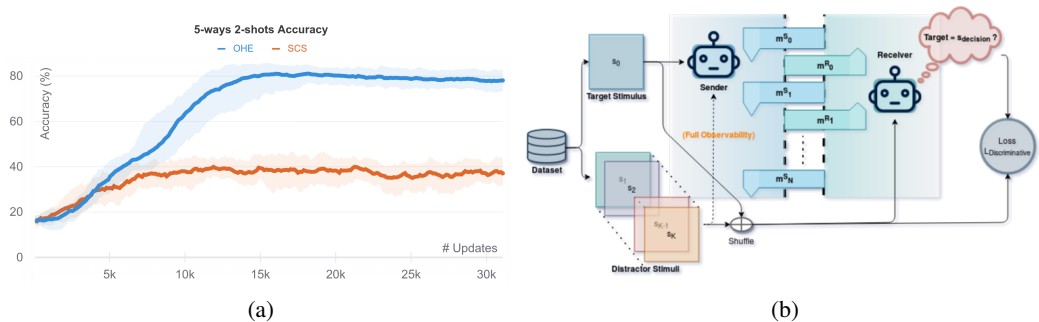

(a)                                        (b)

Figure 2: **(a):** 5-ways 2-shots accuracies on the Recall task with different stimulus representation( OHE:blue ; SCS; orange). **(b):** Illustration of a *discriminative* 2-*players* / *L*-*signal* / *N*-*round* variant of a *referential game*.

### 3.1   OF THE BINDING PROBLEM INSTANTIATED BY THE SCS REPRESENTATION

The SCS representation differs from the OHE one primarily in terms of the binding problem (Greff et al., 2020) that the former instantiates while the latter does not.

More specifically, the OHE representation inherently discloses the semantic structure (i.e. the $d(i)$'s) of the $N_{dim}$-dimensioned symbolic space via the representation shape of the stimuli, i.e. discrete vectors sampled from $\{0, 1\}^{d_{OHE}}$ where $d_{OHE}$ is dependant on the $d(i)$'s. On the other hand, the SCS representation, i.e. continuous vectors sampled from $[-1, +1]^{N_{dim}}$, keeps the semantic structure hidden.

Thus, the semantic structure can only be inferred after observing multiple SCS-represented stimuli. Indeed, we hypothesised that it is via the *dynamic binding of information* extracted from each observations that an estimation of a density distribution over each dimension $i$'s $[-1, +1]$ range can be performed. And, estimating such density distribution is tantamount to estimating the number of likely gaussian distributions that partitions each dimension $i$'s associated $[-1, +1]$ range.

Towards highlighting that there is a binding problem taking place, we show results of baseline RL agents evaluated on a simple recall task. The Recall task structure borrows from few-shot learning tasks as it presents over 2 shots an entire symbolic space.

Each shot consists of a series of games, one for each stimulus that can be sampled from an $N_{dim} = 3$-dimensioned symbolic space. The semantic structure $(d(i))_{i \in [1;N_{dim}]}$ of the symbolic space is randomly sampled at the beginning of each episode, i.e. for each $i$, $d(i) \sim \mathcal{U}(2; 5)$, where $\mathcal{U}(2; 5)$ is the uniform discrete distribution over the integers in the range $[2; 5]$.

Each game consists of two turns: in the first turn, a stimulus is presented to the RL agent, and only a *no-operation* (NO-OP) action is made available to the RL agent, while, on the second turn, the agent is asked to recall the **discrete** $l(i)$ **latent value** that the previously-presented stimulus had instantiated, on a given $i$-th dimension, where the current game's $i$ is uniformly sampled from $\mathcal{U}(1; N_{dim})$ at the beginning of each game. On the second turn, the agent's available action space now consists of discrete actions over the range $[1; max_j d(j)]$, where $max_j d(j)$ is a hyperparameter of the task representing the maximum number of latent values for any factor dimension.

In our experiments, $max_j d(j) = 5$. While the agent is rewarded at each game for recalling correctly, we only focus on the performance over the games of the second shot, that is to say on the games where the agent has acquired theoretically enough information to infer the density distribution over each dimension $i$'s $[-1, +1]$ range, because observing the whole symbolic space once (on the first shot) is sufficient (but not necessary, especially as seen in the case of the OHE representations).

Our results in Figure 2a present a large gap of performance in terms of accuracy over all the games of the second shot, depending on whether the recall task is evaluated using OHE or SCS representations. We attribute the poor performance in the SCS context to instantiation of the binding problem.

# 4 META-REINFORCEMENT LEARNING WITH REFERENTIAL GAMES

## 4.1 REFERENTIAL GAMES

The first instance of an environment that demonstrated a primary focus on the objective of communicating efficiently is the *signaling game* or *referential game* by Lewis (1969), where a speaker agent is asked to send a message to the listener agent, based on the *state/stimulus* of the world that it observed. The listener agent then acts upon the observation of the message by choosing one of the *actions* available to it. Both players goals are aligned (it features *pure coordination/common interests*), with the aim of performing the 'best' *action* given the observed *state*, where the notion of 'best' *action* is defined by the goal/interests common to both players.

Under the nomenclature presented in Denamganaï & Walker (2020a), our benchmark instantiates a *discriminative fully-observable / 2-players / $L = 1$-signal / $N = N_{dim}$-round / uniformly-distributed-$K = 3$-distractors / object-centric* variant. Figure 2b illustrates this setup in the general case.

**Full vs. Partial Observability.** This feature characterises whether the stimuli that are presented to the speaker agent consist of all the stimuli experienced by the listener agent or solely of the target stimulus. For simplicity, and in order for both agents to have the same state space, we employ *full observability* in this benchmark. It also simplify the problem as it allows the speaker agent to reason pragmatically. Specifically, in Figure 2b, the orange arrow highlights the additional information available when the speaker agent has full observability.

**Variable-length Communication** This feature characterises the ability from the speaker agent to send/utter more than one *symbol/signal* to the listener agent, up to a maximal possible length, $L$, for the sequence of symbols. The basic *referential game* is 1-*signalled*. Variable-length communication channels were first introduced by Havrylov & Titov (2017) and has quickly been adopted by the research community as standard, independently of what approach is used to support the communication channel (Lazaridou et al., 2018; Choi et al., 2018). In this work, in order for the action space to remain manageable, we use $L = 1$, and allow for $N = N_{dim}$ communication rounds.

**Multi-Round Communication.** This feature characterises (i) whether the listener agent can send messages back to the speaker agent and (ii) how many communication rounds can be expected before the listener agent is finally tasked to discriminate between the stimuli it observes and have to act by pointing at the one it estimates as the target stimulus. In our proposed benchmark, this feature is parameterizable, but, in this work, in order for both agents to have the same state and action spaces, we allow the listener to send messages similarly to the speaker, but the environment zeros out those messages coming from the listener.

**Stimulus vs. Object Centricism.** The basic (discriminative) *referential game* is stimulus-centric, which assumes that both agents would be somehow embodied in the same body, and they are tasked to discriminate between given stimuli. On the other hand, the object-centric variant incorporates the issues that stem from the difference of embodiment. The agents are tasked with discriminating between objects (or scenes) independently of the viewpoint from which they may experience them. In this variant, the game is more about bridging the gap between each other's cognition rather than (just) finding a common language. It was introduced in the work of Choi et al. (2018), in its descriptive-only form. Needless to say that the object-focused variant adds difficulty to the task. It has been highlighted that embodiment may hold some key to the systematic generalisation abilities of deep learning agents (Hill et al., 2019a), and therefore it is highlighted as a very important research direction to pursue.

In an even more abstract approach, the object-focused setting could be acknowledged as an emphasis on the concept or semantic meaning behind the observed stimulus, and the listener agent would thus be tasked with learning the semantic, while being prompted with different instances of it. In the current work, as we are presenting baseline results, we employ a stimulus-centric parameterisation, but the benchmark we propose incorporates object-centrism and it will be investigated in subsequent works.

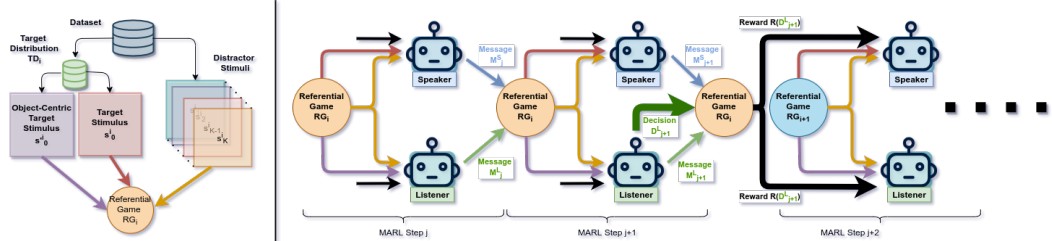

Figure 3: Left: Instantiation of a dataset of SCS-represented stimuli sampled from the current episode's symbolic space, whose semantic structure is sampled out of the meta-distribution of available semantic structure over $N_{dim}$-dimensioned symbolic spaces. Right: illustration of the resulting (meta-)reinforcement learning episode consisting of a series of referential games.

## 4.2 META-REFERENTIAL GAMES

Thanks to the *representation shape invariance property* of the SCS representation, after fixing a number of latent/factor dimension $N_{dim}$, we can define many differently semantically structured $N_{dim}$-dimensioned symbol spaces. In other words that are more akin to the meta-learning field, we can define a distribution over many kind of tasks, where each task instantiates a different semantic structure, that we want our agent to be able to adapt to. And, by defining the tasks as zero-shot compositional learning test that are parameterized by differently semantically structured $N_{dim}$-dimensioned symbolic spaces, we aim for the agent to *learn to exhibit compositional learning behaviours* over $N_{dim}$-dimensioned symbolic spaces.

Figure 3 highlights the structure of an episode, and its reliance on differently semantically structured $N_{dim}$-dimensioned symbolic spaces. Similarly to the Recall task, this meta-referential game-based benchmark presents over 2 shots all the training-purposed stimuli of the current episode's symbolic space, and then over only one shot all the testing-purposed stimuli, similarly to how a *zero-shot compositional test* would be performed with referential games. Each shot consists of a series of referential games, one for each relevant stimulus.

### 4.2.1 VOCABULARY PERMUTATION ON THE COMMUNICATION CHANNEL

We bring the readers attention on the fact that simply changing the semantic structure of the $N_{dim}$-dimensioned symbolic space, is not sufficient to draw out MARL agents adaptation. Indeed, they can learn to cheat by relying on an episode/task-invariant (and therefore semantic structure invariant) emerging language which would encode the continuous values of the SCS representation like a analog-to-digital converter would. This cheating language would consist of mapping a fine-enough partition of the $[-1, +1]$ range onto the vocabulary in a bijective fashion.

For instance, for a vocabulary size $\|V\| = 10$, each symbol can be unequivocally mapped onto $\frac{2}{10}$-th increments over $[-1, +1]$, and, by communicating $N_{dim}$ symbols (assuming $N_{dim} \leq L$), the speaker agents can communicate to the listener the (digitized) continuous value on each dimension $i$ of the SCS-represented stimulus. If $max_j d(j) \leq \|V\|$ then the cheating language is expressive-enough for the speaker agent to digitize all possible stimulus without solving the binding problem, i.e. without inferring the semantic structure. Similarly, it is expressive-enough for the listener agent to convert the spoken utterances to continuous/analog-like values over the $[-1, +1]$ range, thus enabling the listener agent to skirt the binding problem when trying to discriminate the target stimulus from the different stimuli it observes.

Therefore, in order to guard the MARL agents from making a cheating language emerge, we employ a vocabulary permutation scheme (Cope & Schoots, 2021) that samples at the beginning of each episode/task a random permutation of the vocabulary symbols. This approach is similar to the Other-Play algorithm from Hu et al. (2020).

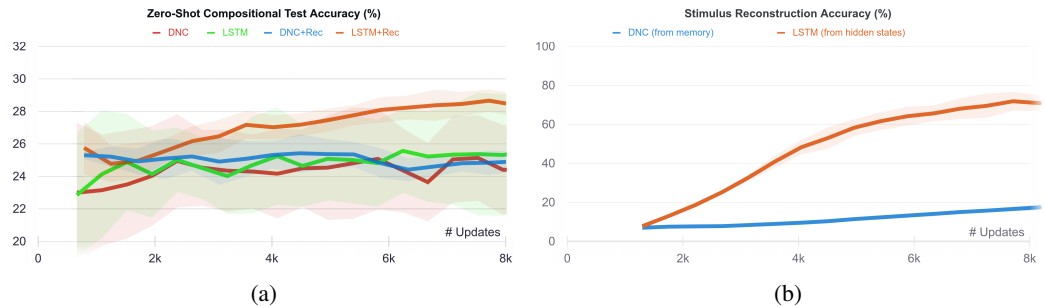

Figure 4: **(a):** 4-ways (3 distractors) zero-shot compositional test accuracies of different architectures. 5 seeds for architectures with DNC and LSTM, and 2 seeds for runs with DNC+Rec and LSTM+Rec, where the auxiliary reconstruction loss is used. **(b):** Stimulus reconstruction accuracies for the archiecture augmented with the auxiliary reconstruction task. Accuracies are computed on binary values corresponding to each stimulus' latent dimension's reconstructed value being close enough to the ground truth value, with a threshold of $0.05$.

## 5 EXPERIMENTS & ANALYSIS

### 5.1 SINGLE-AGENT REINFORCEMENT LEARNING SETTING

While a referential game usually involves a speaker and listener agent, in the present paper we propose to focus solely on the listener agent by replacing the speaker agent with a rule-based speaker agent whose language is (linguistically) compositional in the sense of the **posdis** compositionality metric (Chaabouni et al., 2020).

Indeed, the listener agent of a referential game is solely focused with the problem of language acquisition, which is assumed easier than the problem of language emergence that the speaker agent has to solve by searching for an expressive enough artificial language to describe SCS-represented stimuli.

By focusing solely on the language acquisition problem, in the context of a (linguistically) compositional language, we focus in effect on the listener agent's ability to learn compositional learning behaviours, and nothing more. We will explore the other problems of this framework in subsequent works.

### 5.2 AGENT ARCHITECTURE

The baseline RL agents that we consider are made up of standard architecture for reinforcement learning in multi-modal (stimulus + language) environments. The stimulus is processed at every timestep by a fully-connected 3-layers network. The language input, represented as a one-hot-encoding, is concatenated with the stimulus embeddings and, finally, a core memory processes the information over time. A fully-connected layer followed by a softmax activation maps the output of this core memory module to a distribution over 29 actions, which corresponds to an action space combining both the language output and the decision output. Optimization is performed via an R2D2 algorithm(Kapturowski et al., 2018).

We investigate both an LSTM (Hochreiter & Schmidhuber, 1997) and a Differentiable Neural Computer (DNC) (Graves et al., 2016) as core memory module. More details can be found, for reproducibility purposes, in our open-source implementation at HIDDEN-FOR-REVIEW-PURPOSE. Hyperparameters have been selected via fine-tuning using Weights&Biases' Hyperparemeter Sweep feature.

### 5.3 AUXILIARY RECONSTRUCTION TASK

Following the work of Hill et al. (2020), we augment the agent with an auxiliary reconstruction task aiming to help the agent learning to use its core memory module. The reconstruction loss consists of a mean squared-error between the stimuli observed by the agent at a given time step and the

output of a prediction network which takes as input the current state of the core memory module after processing the current timestep stimuli. In the case of the LSTM, the hidden states are used, while in the case of the DNC, the memory is used as input to the prediction network.

## 5.4 RESULTS

Figure 4a shows the 4-ways (3 distractors) zero-shot compositional (ZSC) test accuracies of the different agents throughout learning. The zero-shot compositional test accuracy is the accuracy over testing-purpose stimuli only, after the agent has observed for two consecutive times the supportive training-purpose stimuli for the current episode/task parameterised by the current semantic structure. The DNC-based architecture has difficulty learning how to use its memory, even with the use of the auxiliary reconstruction loss, and therefore it utterly fails to reach better-than-chance zsc test accuracies. On the otherhand, the LSTM-based architecture is fairly successful on the auxiliary reconstruction task, but it is not sufficient for training to really take-off. This result hints at the fact that indeed new inductive biases must be investigated to be able to solve the problem posed by the benchmark that we propose.

## 6 CONCLUSION

In order to build artificial agents able to collaborate with human beings, we have proposed a novel benchmark to investigate artificial agents abilities at learning compositional learning behaviours. Our proposed benchmark casts the problem of learning compositional learning behaviours as a (meta-)reinforcement learning problem, using iterated (discriminative) referential games, containing at its core an instantiation of the binding problem. This instantiation of the binding problem and the meta-learning formulation of referential games is made possible by a novel representation shceme, entitled the Symbolic Continuous Stimulus (SCS) representation, which acts as a versatile representation for symbolic spaces as it allows for infinitely many semantic structure to be instantiated without changing the shape of the representation, and it relies on continuous values that are more akin to real world stimuli, as opposed to discrete valued one-hot-encodings.

Finally, we have provided baseline results on the single-agent tasks of learning compositional learning behaviours, using state-of-the-art RL agents built around core memory modules, and our results show that our proposed benchmark is currently out of reach for. Thus, we hope will spur the research community towards developing more capable artificial agents.

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
