# OpenReview forum: "Meta-Referential Games to Learn Compositional Learning Behaviours"
_ICLR.cc/2022/Conference — ICLR 2022 Submitted_

### Official Review · Reviewer_VGZD · 2021-10-25

**Correctness:** 1
**Technical Novelty And Significance:** 2
**Empirical Novelty And Significance:** 2
**Recommendation:** 1
**Confidence:** 4

**Main Review:**

This is an ambitious paper that could offer some promise. Learning to generalize compositionally is an intriguing idea. However, I think the work needs substantially more development along a number of axes:

SCS vs. purely continuous representations
---------------

This may partly be an issue of communication (see below), but it's not entirely clear to me why the authors don't consider fully continuous representations in addition to OHE or SCS ones. The argunments of prior work referenced below (e.g. Chalmers, 1990) show that continuous representations can be effective for compositional generalization, as do many of the empirical studies the authors exhibit in RL, as well as the successes of large language models, etc. All modern approaches rely on continuous representations, and often achieve some degree of effective compositional generalization. It certainly seems important to run a continuous-representation comparison condition to justify the value of the SCS encoding.

Indeed, the authors (accurately) criticize OHE for not being able to extend without changing the shape of the vector representations. However, their SCS representation is subject to limitations as well, so far as I can see: it cannot adapt to more latent dimensions than are pre-allocated, and I suspect that it also would fail to extrapolate to dimensions with a larger $d(i)$ than the architecture saw in training (although this is speculation, and perhaps something that could/should be evaluated experimentally in a revised version of this work).

Furthermore, constructing an SCS seems to require a) that the semantics of the task be simple enough to be stated as discrete values along a small number of latent dimensions, and b) that we be able to construct these representations from the inputs (which might generally be e.g. images). Indeed, continuous representations are used in basically all modern applications for just this reason---we don't know how to construct an appropriate discrete representation for the space, and it's not clear if one even necessarily exists. Thus, in order for this method to be applicable beyond toy settings, it would be useful for the paper to explain how this might work. Ideally, the authors would *demonstrate* such success on existing benchmarks for compositional generalization (see below).

Clarity
------------

Neither the paradigm nor the results are stated in sufficient detail or clarity to understand how to build upon the work. For example:

* The rule-based speaker the authors use to evaluate uses compositional language, but what exact form does this take? This should be clearly stated, since it is central to understanding the main experimental evaluations.
* The only description of how the agent's take communication actions is that the output is "a distribution over 29 actions, which corresponds to an action space combining both the language output and the decision output." How are these combined?

These kind of details can make a substantial difference in RL, and should certainly be reported. Furthermore, there are many other clarity issues, e.g.:

* The paper spends a great deal of time discussing issues like "object-centric" representations vs. "stimulus-centric" ones, but I do not see how such notions could apply to the SCS representation as suggested by the paper. Or are the stimuli represented differently as inputs to the agents? Even this detail of the experimental evaluation is not clearly stated.
* What the paper calls a "One-Hot Encoding" OHE seems to really be a multi-hot encoding (that is, it is one-hot along each dimension, but a single stimulus has multiple  1s within it). However, this is not clearly explained, and it's only really communicated if the reader carefully examines figure 1. A point like this should be clarified in the text.


Evaluation
------------

The results of this approach are barely-above chance performance (29% at most, where chance is 25%). While this may be interesting in some sense, it is not a particularly compelling endorsement of the SCS representation.

In addition, even this experimental evaluation is limited. First, the results suggest poor tuning of the  hyperparameters: the DNC performs worse than an LSTM memory, but a DNC *contains* an LSTM in the controller, and so should almost certainly perform better (even if just by learning to ignore the memory part). Of course, these are challenging architectures to optimize, but this means that the claims of the paper are suspect: perhaps the performance is low for reasons of poor training, rather than inherent difficult of the task. Furthermore, the authors perform effectively no ablation studies, or further evaluation of which aspects of the approach matter. All of these would be necessary to make a valuable contribution to the literature.
Finally, it would be much more interesting if the authors could demonstrate the value of their approach on existing compositional generalization challenges such as SCAN or gSCAN (which they cite), where baselines do exist. Could SCS offer benefits in these more complex settings? This would force the authors to grapple with some of the challenges outlined above about using SCS in environments with slightly more complexity (but still much simpler and more amenable to structured representations than the stimuli used for reference games in cognitive psychology).

Literature
-----------

As the authors note, compositional generalization has been a major topic of research for some time. The work needs to be better situated within this broader literature, to clarify its contributions. For example, the author's points about discrete vs. continuous representations overlap with long-standing replies to Fodor & Pylyshyn's claims about compositionality. For example, see Chalmers (1990) or Smolensky (1987), as well as much of Smolensky's subsequent research (e.g. Smolensky, 1990; McCoy et al., 2018). This paper would convey its contributions much more clearly if it were situated within this broader literature.

I don't think it's as essential to engage with with, but the authors may also be interested in Santoro et al. (2021), an opinion piece which discusses a number of issues related to several aspects of this paper, including discussions of discrete vs. continuous symbol representations, and arguments that communication provides a unique path to instilling symbols.







References
-------------

Chalmers, D. (1990, July). Why Fodor and Pylyshyn were wrong: The simplest refutation. In Proceedings of the Twelfth Annual Conference of the Cognitive Science Society, Cambridge, Mass (pp. 340-347).

Santoro, A., et al. (2021). Symbolic behaviour in artificial intelligence. arXiv preprint arXiv:2102.03406.

Smolensky, P. (1987). The constituent structure of connectionist mental states: A reply to Fodor and Pylyshyn. Southern Journal of Philosophy, 26, 137-161.

Smolensky, P. (1990). Tensor product variable binding and the representation of symbolic structures in connectionist systems. Artificial intelligence, 46(1-2), 159-216.

McCoy, R. T., Linzen, T., Dunbar, E., & Smolensky, P. (2018). RNNs implicitly implement tensor product representations. arXiv preprint arXiv:1812.08718.

**Summary Of The Paper:**

This paper proposes to investigate meta-learning to generalize compositionally. This is an important challenge. The paper proposes a meta-learning setting consisting of reference games, which are a classic paradigm for investigating communication. The paper proposes a hybrid symbolic-continuous stimulus representation, and explores how different agents use this representation scheme within their paradigm.

**Summary Of The Review:**

This is an ambitious paper that could offer some promise. Some of the ideas are intriguing. However, to achieve its potential, this paper needs substantially more development conceptually, in terms of evaluation, in clarity of communication, and in engagement with the existing literature.

---

### Official Review · Reviewer_3vEd · 2021-11-02

**Correctness:** 3
**Technical Novelty And Significance:** 2
**Empirical Novelty And Significance:** 2
**Recommendation:** 3
**Confidence:** 3

**Main Review:**

**SCS**

The authors propose a new method to represent symbolic stimuli as continuous representations as an alternative to one-hot encodings. The SCS representations seem like a flexible way to define differently structured semantic spaces in synthetic settings (one might need to change the dimensions of the input if they were to use one-hot encodings). The paper then does a good job of showing how using one-hot encodings compared to SCS change the problem definition leading to a difference in performance on the same task.

Despite this, I fail to see how the SCS representations might be useful outside the scope of the experiments shown in the paper. I also encourage the authors to simplify the experiment described in section 3.1 to make it more clear. The applicability and the novelty of the SCS representation seem limited. I would like the authors to have a more extended discussion of how SCS can be used outside of their work.

**BENCHMARK & BASELINES**

The paper does an excellent job of introducing the meta-referential games and the different setup settings they use, and the corresponding reason behind them.

While the authors say that they use vocabulary permutation, it is not clear what the size of the vocabulary is. Similarly, I could not find the value of the dimension of the symbolic space $N_{dim}$ that was used to generate the stimuli. In a similar vein, I would have appreciated more details about the model architecture and the training regimes used (possibly in the appendix).

I found the result that the LSTM-based model successfully reconstructs the stimuli but fails on the main game to be interesting. But without additional details about the failure or examples of failure cases and a discussion about what these results imply outside the tasks they were tested for, I was not convinced by their utility. I would also urge the authors to have a speculative discussion on what successful inductive biases might look like.

**Other Comments:**

Page 2: Further more → Furthermore

Page 3: On-hot-encoded → One-hot-encoded

Page 4: Relational Frame Theory (RTF) →Relational Frame Theory (RFT)

Page 6: Simplify → Simplifies

Page 6: observes and have → observes and has

Page 9: zsc → ZSC

Some lines in the introduction and the related work sections are incredibly similar. Eg:

The following line is exactly the same in the Introduction and Related Works section.

Introduction and in Related Woks: "Chaabouni et al. (2020) showed that, when a specific kind of compositionality is found in the emerging languages (the kind that scores high on the positional disentanglement (posdis) metric for compositionality that they proposed), then it is a sufficient condition for systematicity to emerge."
I would advise the authors to make the introduction and related works sections more succinct.

**Summary Of The Paper:**

The paper presents a new benchmark based on a referential communication game (where a speaker has to communicate the stimuli they observe to the listener and the listener has to find the right stimuli among distractors) to test for compositional learning behaviors. The stimuli vary over differently structured semantic spaces (every dimension can have different numbers of semantic types). Models are trained on a series of meta-referential games, where each game has a different semantic distribution structure (different number of semantic types) and are evaluated on zero-shot performance on a referential game with a new semantic structure. To represent the stimuli, the authors propose a continuous vector representation for different referents instead of using a one-hot embedding. Finally, the authors show how commonly used deep learning architectures fail on their benchmark, posing a challenge to the community.

**Summary Of The Review:**

I wasn't convinced of the utility of the three main contributions of the paper outside the limited setting described in the paper: SCS, the meta-referential game and the failure of models on the same. Further, the paper is unclear in several places and does not provide enough details. Thus, in its current form, I recommend that the paper be rejected.

---

### Official Review · Reviewer_o5wV · 2021-11-02

**Correctness:** 3
**Technical Novelty And Significance:** 2
**Empirical Novelty And Significance:** 2
**Recommendation:** 3
**Confidence:** 3

**Main Review:**

I think the posed problem, using meta-RL for continual referential games with different semantic structures, is an interesting problem. However, the paper is difficult to follow, and details are missing from models and experiments.

1- The paper is difficult to follow; especially the introduction where the authors give a lot of related work without a clear introduction of the problem and why it is important. I think it would help if the introduction is kept simple and references are moved to related work.

- Upon first reading the abstract, I thought one of the main contributions of the work is a new benchmark for compositional learning. But, it is only mentioned on the last paragraph of the introduction without much explanation on what it entails. I think there needs to be more emphasis on what the benchmark is; how it is different from existing compositional benchmark, what it is measuring, how it is structured etc.

- There are several concepts that the paper uses but are never defined such as "positional disentanglement metric" (where the speaker uses a language with this characteristic) or "systematic generalization" (that the paper is targeting).

- I think more concrete examples are needed to make the paper more readable such as a real stimulus example (for Figure 1), sample conversations between two agents to understand compositional learning, sample tasks and meta-tasks, etc.

- Sentences are in general long and repetitive in some cases such as the last sentence of abstract or "differently semantically structured symbolic spaces".

2- I think more discussion on why OHE shouldn't be used and SCS should be preferred is needed. Stimuli are already discrete (Figure 1) and should come from a bounded vocabulary, nonetheless a very large one, why would using OHE be prohibitive for meta-learning? Considering that humans communicate with a finite vocabulary and can generate unlimited compositional structures, why can't we use a similar approach to represent stimuli? I think a more concrete example that shows a realistic continual task learning problem with discrete stimuli where the boundary of the stimuli can't be fixed and needs to be significantly increased over time is needed.

- Your initial experimental results in Figure-2 also suggest that OHE clearly gives higher performance and SCS might be introducing unnecessary noise to the process.
- It is also not clear to me whether the performance of SCS in Figure-2 can be improved or not. For example, can we use millions of updates to achieve results on par with OHE? Instead of sampling d(i) randomly, if you equally divide the interval [-1,1] into two sections and take N_{dim}=2, what would be the performance?

3- For the experiments in Section 3.1., there is no setup. I think a more detailed setup that includes what is the baseline RL agent, how it is trained in this 2-player game, what is its architecture, how many conversations took place, etc.

4- The setup for meta-referential games using SCS is not clear to me. If at any point in the future, we introduce more stimuli which increases the boundary, causing problems with OHE, this means that we are dividing the interval [-1, +1] into more and more sections. Wouldn't this cause intersections with previous SCS representations and lead to ambiguity? Are you suggesting that, over time the agents would forget some of the earlier tasks that they learned to accommodate for new tasks?
- I think some ablations are needed here, including increasing the boundary of the stimuli gradually for new tasks to see learning progress (and forgetting), increasing number of latent dimensions N_{dim}.

5- In Figure 4 (a), there is no improvement other than LSTM+Rec. This is strange to me as using only ~1K updates gives the same performance as using 8K updates. Could you explain the reason why?

6- Could you explain what you mean by "supportive training-purpose stimuli" in Section 5.4? Do you mean these are supportive for the test task adaptation?

7- Some grammatical mistakes:
- Apostrophes are missing in some places, "Neural networks' induction biases", "Both players' goals".
- "Further more" --> "Furthermore"
- "On the otherhand" --> "On the other hand"
-"shceme" --> "scheme"
- "we hope will spur" --> "we hope it will spur"

**Summary Of The Paper:**

This paper studies the problem of learning compositional behaviors through referential games. The authors first introduce the problem of fixed shape representations that are not suited for continual learning with infinitely many tasks. They propose a continuous representation of discrete stimulus by discretizing a small interval and randomly creating gaussian distributions at each interval. They then define the compositional learning problem using meta-reinforcement learning where a set of new tasks are introduced over time and two agents communicate to develop a compositional language. The authors present some empirical results where DNC and LSTM based agents are not well tailored to solve the problem and exhibits very low accuracy while the LSTM based agent is able to reconstruct an input stimulus successfully.

**Summary Of The Review:**

I think the overall problem of compositional learning through meta-referential games is interesting but the paper is hard to follow and its main contributions should be emphasized more; especially the benchmark should be detailed with more results, ablations and the new representation, SCS, should be clarified.

---

### Official Review · Reviewer_RCbW · 2021-11-03

**Correctness:** 3
**Technical Novelty And Significance:** 2
**Empirical Novelty And Significance:** 2
**Recommendation:** 3
**Confidence:** 2

**Main Review:**

The authors introduce a symbolic continuous stimulus representation that has the particularity of enabling the representation of stimuli sampled from differently semantically structured symbolic spaces while maintaining the same representation shape.

The paper is a bit hard to read, I am not sure I fully understand this paper. Could the authors provide more details of why the proposed representation is helpful for compositional learning?

Is the proposed method adaptable to solve other tasks that do not belong to preferential games?

The experiments are not sufficient to support the claimed contributions. The authors did not compare the proposed SCS representation with other representations, for example, which baselines use one-hot-encodings?

The text in Figure 1, Figure 2, and Figure 3 are too small.

**Summary Of The Paper:**

In this paper, the authors propose a new benchmark to investigate state-of-the-art artificial agents' abilities to exhibit compositional learning behaviors. The authors propose a Symbolic Continuous Stimulus (SCS) representation and cast the problem of learning compositional behaviors as a meta-reinforcement learning problem. The authors compare state-of-the-art RL methods on solving the single-agent task of learning compositional learning behaviors.



**Summary Of The Review:**

I think the idea of compositional learning is interesting. However, I didn't fully understand why the proposed representation is helpful for compositional learning? I hope to see more explanations from the authors in their feedback.

The experiments are not sufficient as well. The authors simply compared several baselines without extra ablation study of their proposed method. It is hard to know why LSTM+Rec is better than other baselines. The authors just use 2 seeds for runs with DNC+Rec and LSTM+Rec, they should report the results of using more seeds in Figure 4.

I think this paper needs to be improved and it has the potential to become a promising paper.

---

### Author Response · Authors · 2021-11-23
**Thank you**

We thank the reviewers for their insightful reviews: thank you very much!

We will improve the paper, making it easier to understand and easier to build upon, with the reviewers' comments in mind.

---

### Decision · Program_Chairs · 2022-01-20

**Decision:**

Reject

**Comment:**

All reviewers have agreed that the topic of evaluating compositional skills of agents is an important one and cast it as compositional learning as meta-reinforcement learning is an interesting approach. At the same time, reviewers have raised concerns with respect to the benchmark itself, the exposition and clarify of the ideas as well as the experimental evidence used to support some of the claims. The authors have not provided an author response but have acknowledged the reviewers feedback.

As this paper stands I cannot recommend acceptance for the current manuscript.